# Permeability Sensors for Magnetic Steel Structural Health Monitoring

**DOI:** 10.3390/s25030606

**Published:** 2025-01-21

**Authors:** Evangelos V. Hristoforou

**Affiliations:** Laboratory of Electronic Sensors, Zografou Campus, National Technical University of Athens, 15780 Athens, Greece; hristoforou@ece.ntua.gr

**Keywords:** structural health monitoring, magnetic steels, barkhausen noise, hall effect, anisotropic magnetoresistance

## Abstract

In this paper, magnetic permeability sensors able to perform structural health monitoring of magnetic steels, by means of determining residual strain and stress amplitude and gradient distribution, responsible for crack initiation, are presented. The good agreement between magnetic properties and residual strains and stresses is illustrated first, resulting in the determination of the magnetic stress calibration (MASC) curves and the Universal MASC curve. Having determined differential magnetic permeability as a key magnetic property, able to measure and monitor residual strain and stress distribution in magnetic steels, the paper is devoted to the presentation of the permeability instruments and sensors developed in our lab. The classic single sheet testers and the electromagnetic yokes, are compared with new, low-power-consumption permeability sensors using the Hall effect and the anisotropic magnetoresistive (AMR) effect, discussing their advantages and disadvantages in magnetic steel structural health monitoring.

## 1. Introduction

Magnetic steels cover a large range of structural and operational applications. Among them, transportation, energy, metallic buildings and bridges, as well as oil and gas are vital for the society and economy, requiring preventive maintenance. Apart from these, special scientific structures, like the steel tube rings used in the “Conseil Européen pour la Recherche Nucléaire” (CERN), the European Council for Nuclear Research and other important scientific infrastructure require special attention, particularly concerning the employed steel grades. In some cases, like ITER in southern France, stainless steel grades, like 316LN, being non-magnetic, are also used and need preventive maintenance. In general, a large variety of different steel grades are used in different specific applications. The preferable type of steel grade depends on its use: the grade, treatment and thickness of the used steel are dependent on the environment, like chemical ambient atmosphere, load, hydraulic stresses and low–high cycle fatigue, as well as environmental conditions like temperature, humidity and their gradients, in which it is used. The variety of these steels is so large, starting from single ferritic structures, low-carbon compounds, duplex steels, TRIP steels, maritime steels and so many other grades, that it is difficult to implement a universal method for maintenance and especially for preventive maintenance. For this reason, steel producers and manufacturers, like pipeline manufacturers, shipyards, heat exchanger producers, etc., follow specific procedures in alloying and post-manufacturing treatment, allowing for the proper and required structure and microstructure [1].

Steel failure may be critical and vital for several steel structures [2]. Steel failure in the energy cluster may result in disruptive damages. The domes of nuclear stations, suffering neutron fluence and therefore embrittlement, must be secure, far from being damaged, otherwise the public health of close and distant towns would be in danger, like in the Chernobyl accident. Environmental catastrophes may also affect nuclear stations, like in the recent Fukushima nuclear accident. Apart from that, steel failure in renewable, lignite and hydrogenation stations is also of importance, resulting in the disability of energy transmission. Similar problems appear in transportation. Railways and trains, suffering thermo-mechanical fatigue, in either a low or high cycle fatigue process, are in danger especially in large temperature gradients. For example, in north railway lines, sunshine causes elevated temperatures on the railways, so that at the day-to-night transition, the thermal gradient together with the mechanical fatigue may result in railway failure. After the privatization of railways, especially in northern countries with high thermo-mechanical fatigue, where the preventive maintenance depends on the profit of the railway operation, such accidents are rather frequent. Shipping is also a subject of steel failure, in the hull structure, as well as in shaft conditioning. Recently, shaft accidents that impact the ship’s steering and propulsion ability are frequent, illustrating the need of preventive structural health monitoring and maintenance. The new era of electrification also requires special attention, not in terms of steel failure but in magnetic properties performance, requiring frequent operational conditioning monitoring with corresponding maintenance by means of localized magnetic annealing. The sector of oil and gas also requires special attention. The harsh environment of oil extraction, refineries and oil and gas transmission pipelines require special types of steels, which operate in a large variety of chemical environments at elevated temperatures. These conditions may result in unforeseen accidents, such as the Mexico gulf accident. On the other hand, steel structures like skyscrapers, metallic bridges and the like must also be protected against damage. Apart from these applications, the structural and conditional health monitoring in large scientific experiments like in CERN, ITER, etc. must be properly monitored in terms of structural and microstructural characterization. However, all these issues of structure and microstructure in all these steels grades can be summed up in one major parameter, affecting their structural health, namely the residual strains and stresses (named hereinafter stresses for simplicity reasons). These stresses are the accumulation of several different reasons for structural imperfections, like dislocations, precipitations. etc.: steel failure is dependent on the gradient of residual stresses [3]. These residual stresses are also subject to fatigue conditions, namely low and high cycle fatigue, concerning stresses above and below the yield point of the corresponding steel. In fact, laboratory tests under low and high cycle fatigue allow for the determination of the remaining lifetime of the corresponding steel structure. This challenge is rather difficult to be determined, being dependent on the specific steel under investigation and the environmental and stress conditions. However, the problem of the remaining lifetime can be actually solved, by knowing the residual strain and stress amplitude distribution. The processed signals can often be subject to uncertainties and/or inaccuracies. For this reason, recent scientific literature suggests using techniques based on fuzzy logic. In particular, the most recent research guidelines recommend grouping similar data in a fuzzy sense (through fuzzy divergence calculations that prove to be a measure of distance in a particular functional space) into specific groups to extract, for each of them, a representative characteristic of each grouping. Then, the classification of the signal can be performed by comparing the obtained fuzzy divergences with each representative data of each grouping [4].

The reference method to determine and monitor residual stresses is diffraction. The surface residual stresses are detected by X-ray diffraction, preferably following the Bragg–Brentano set-up (XRD-BB) [5], while the bulk residual stresses are detected by neutron diffraction (ND) [6]. The spatial resolution of diffraction can be as low as 1 mm^2^ using a high density of photons or neutrons. The methodology of the measurement of stresses is analyzed in the next chapter. Apart from these two classical methods, other techniques are also available, like synchrotron, offering even higher spatial precision. The currently used stress sensors in the field are strain gauges [7], while the drill hole method is also an alternative [8]. Strain gauges measure the local strain change after they are fixed in position. Their advantage is the straightforward measurement of strain changes, while the laborious methodology of fixing them on the steel surface, as well as their disability to monitor the distribution of strains, are their main disadvantages. Drill hole is a standardized method to monitor the stress tensor: after drilling a circular well on the steel surface, the circle is arranged to an ellipse, where the long and short diameters indicate the tensile and compressive stress history, respectively. Although the drill hole method is able to determine the history of stresses, it can only be used locally. The method is also somehow destructive.

For all these reasons, there is a great interest in determining and developing a non-destructive and contactless, if possible, method to provide stress and strain distribution measurement and monitoring along the surface and the bulk of steels. Such a method can be magnetic: it can be contactless and certainly non-destructive with the ability of scanning in several cases. Magnetic sensors can also be used in harsh environments, provided that the required power consumption permits their use. Our lab has studied such magnetic methods to correlate magnetic properties with residual stresses in magnetic steels for more than three decades. Apart from that, we have provided three different families of magnetic sensors, namely magnetic Barkhausen noise [9], differential permeability [10] and magnetoelastic sensors [11]. In this paper, after summarizing our methodology concerning the correlation of magnetic properties with residual stresses, we emphasize on the permeability sensors, namely the single sheet testers and the electromagnetic yokes, that serve as reference tools and lab equipment to determine stresses, as well as low consumption permeability sensors, like Hall and AMR sensors for in-field applications and measurements.

## 2. Magnetic Stress Calibration Curves

The use of any kind of instrument to monitor magnetic properties for the determination of residual stresses requires the comparison of residual stress and the magnetic property at the very same area-volume and direction, resulting in the determination of sensitivity, uncertainty, parametric dependence of uncertainty as well as the spatial resolution of the magnetic method. For this reason, the development and maintenance of magnetic steel samples, called hereinafter steel coupons, are necessarily required with well determined residual stresses on their surface and in their bulk.

Our group has initially taken the advantage of welding techniques to develop such stress coupons for calibration reasons. In fact, using autogenous steel welding of the same steel grade, it is possible to generate a steel coupon with a stress gradient across the weld. (Autogenous welding means welding without any other additional compound to be added in the steel under welding.) For simplicity reasons, the stress coupons are usually flat. Such autogenous welding methods that can generate a stress gradient across the welding are the tungsten inert gas (TIG) method [12], the plasma welding method [13], the induction heating method [14], the electron beam welding [15], etc. The first three mentioned welding methods are widely used in the steel industry, while the fourth one is mainly used for research and development purposes. In all these welding techniques, the welding method is the same: the two parts of steel are molten and then pressed to be in touch or even mix each other. This way, during cooling, a stress gradient can be generated in the fusion zone (FZ) and the heat-affected zones (HAZ), provided that the two steel parts are forced to be flat. Practically, the base metal (BM) remains with a minimum amount of stresses. Otherwise, if the two components are not forced to be flat, the cooling welded steel is deformed to accumulate the heat gradient release.

Figure 1a illustrates such residual stress distribution in the fusion zone and heat-affected zones. Despite the phase transformations introduced inside the welded steel, the diffraction methods are capable of determining the residual strains existing within the FZ and HAZ areas [16]. All our steel coupons until now are based on such welding methods, despite issues like phase transformations and repeatability process. However, both phase transformation and sample repeatability are important and enlarge the uncertainty in inter-laboratory comparison tests. For this reason, we have started using another method, according to which phase transformation and sample repeatability are much better, allowing safer inter-laboratory comparison tests. This method is the local induction heating using a 6 mm straight, water-cooled conductor, transmitting a high-frequency current, generating eddy currents and therefore heat along the steel. The heated steel is set on top of a ceramic base and two vices are loosely fixing the ceramic and the steel under heat. Due to the relatively low thermal conductivity of steels, a temperature profile is generated in the steel, perpendicular to the orientation of the straight conductor direction, which can be regarded as a heat-affected zone (HAZ). The temperature distribution of the steel is monitored by an infrared camera. As soon as the maximum temperature reaches temperatures in the order of one-third of the melting temperature of the steel grade, thus allowing for no phase transformations, the two vices fix the ceramic-steel assembly and the whole assembly is inserted either in cold water or oil for quenching. This way, the temperature gradient is transferred into residual stresses, until the temperature in the steel is uniform (Figure 1b).

The steel coupons used for the experiments of this paper are both welded and heated-quenched ones. These steel coupons must be kept in a safe place, without temperature, humidity and stress. Practically, they are stored in zero humidity champers, covered by soft blanquettes to avoid changes in localized residual stresses. Steel coupons of homogeneous stresses could also be prepared and stored, following heating, consequent quenching and then thermal recovery with different temperature and time intervals. However, this is not practically useful for in-field calibration reasons and therefore it is not followed as a best-practice. Then, these steel coupons undergo stress characterization. Thin, white indelible dots are used to mark different positions on top of the quenched steel, across the heat-affected zone, where the measurement of residual stresses is realized. Three lines of dots are designated on the surface of the steel, to have a number of areas with residual stresses for statistical reasons. They are determined in given directions, namely X and Y directions for surface stresses and X, Y and Z directions for bulk stresses. The surface strain measurement using the XRD-BB arrangement will be illustrated hereinafter. Pointing in one (let us say X) direction on an individual dot point of the steel coupon, the XRD response is determined, selecting the most stable 2theta angle with respect to sample tilting. We have experienced that the fourth peak of the XRD of ferrous steels is the most stable, which is close to 98°–99°. Then, the X-ray source and detector are fixed in this angle with a fourth decimal precision and rotate in an angle ±ψ from the vertical line, from −35° to +35° with respect to the sample, monitoring the change of the detected counts. Thus, the dependence of counts on ${sin}^{2}\psi$ determines the strain ε in this direction at the given dot, by the following formula:(1)$$\varepsilon_{x}=\frac{\frac{D_{1}-D_{n}}{D_{n}}}{(1+n){sin}^{2}\psi }$$

Positive inclination of this dependence corresponds to tensile stresses and negative inclination to compressive stresses. Then, the Young’s modulus Ε, determined by the stress–strain curve of the steel, multiplied with ε_x_, results in the residual stress $\sigma_{x}$:(2)$$\sigma_{x}=Ε\frac{\frac{D_{1}-D_{n}}{D_{n}}}{(1+n){sin}^{2}\psi }$$

This process is repeated in the Y-axis and then, the two stress components of this dot are determined. Τhe next dot measurement follows, until the strains and stresses in all directions and dots of the steel are determined. The precision of the 2theta angle determination, the type of the X-ray source and the precision of the counts detected in different angles are the three main and decisive parameters for the quality of the dependence of counts on ${sin}^{2}\psi$. In case one of these parameters is not determined well, the dependence of counts on ${sin}^{2}\psi$ is not monotonic, resulting in large uncertainties in stress measurement. The counts are detected with a signal to noise ratio of 10^4^ due to the time exposure in photons All this process is depicted in Figure 2. Neutron diffraction can detect the strains and stresses in the bulk of the steel, following a very similar procedure.

In all measurements of residual stresses, the initial measurement was the microstrain change, which was always multiplied with the Young’s modulus E of the original compound, without taking into account the phase transformations or microstructural changes. This practically means that all measurements and experiments referred to microstrain changes. This ensures the validity of the measurements, avoiding any change of the Young’s modulus E or any phase transformation and microstructural change.

Magnetic properties are measured on the surface and in the bulk of these steel coupons. These magnetic properties should be related to magnetic Barkhausen noise, magnetic permeability and magnetoelastic properties of the material, i.e., the dependence of the magnetostriction coefficient λ on residual stresses. These properties can be determined by measuring Barkhausen noise, magnetization hysteresis and magnetoacoustic emission measurements. Barkhausen noise can determine the amount of Barkhausen jumps by counting the number of pulses at each period of excitation, as well as the energy needed to overcome the obstacles of domain wall propagation, which is proportional to the integral of the Barkhausen jumps with respect to time. In practice, this energy is represented by the surface below the envelope of Barkhausen jumps. Magnetization loop offers the differential permeability dependence of the applied field and can be partially considered as the integral of the Barkhausen jumps sequence. In fact, the maximum differential permeability is the preferred magnetic property of the magnetization loop, being more sensitive than others, like coercivity, remanence magnetization and magnetic losses. The dependence of static magnetoelastic properties, namely the dependence of λ on stress, as well as the dependence of dynamic magnetoelastic properties, namely the amplitude and waveform of magnetoelastic waves generated in magnetic steels, that also offer significant information on stresses, is out of the scope of this paper.

We have initiated the measurements of Barkhausen noise and differential permeability using the MEB-2c probe of the Magnetics Laboratory, Gdansk, Poland (Figure 3). This probe uses an electromagnetic yoke to magnetize the steel under test, a search coil on the electromagnetic yoke to obtain a response proportional to the differential permeability of the steel under test, as well as a magnetic Barkhausen noise (MBN) search stylus set on the surface of the steel under test to provide the Barkhausen noise characteristics. The MBN stylus comprises of a short length and diameter coil, containing a thin ferrite rod for amplification of the MBN signal, which is weak and requires some magnetic and electronic pre-amplification.

Having had the tools to determine and monitor magnetic properties, the correlation between stresses and magnetic properties was realized, by means of measuring MBN and differential permeability at the same areas and directions, where residual stresses were measured (Figure 4). The reasons for good agreement are mainly related to the correlation of magnetic properties and stresses in the same direction, using a proper X-ray source and magnetic instrument. These measurements indicate that the MBN measurements are in good agreement with residual stresses. The output of the search coil related to the differential permeability measurements indicate less precise measurements, since the yoke was not calibrated, as described next in this paper.

From these results, the pairs of stress and magnetic measurements resulted in the so-named magnetic stress calibration (MASC) curve. Until now, we have achieved 26 different types of MASC curves from different steel grades. All these curves are different to each other, in terms of residual stress amplitudes due to the corresponding yield point, as well as in magnetic properties due to the different MBN and permeability response. However, normalizing the X-axis with the corresponding yield point of the steel grade and the Y-axis with the maximum magnetic property (MBN or permeability), all MASC curves collapsed in on MASC curve, called Universal MASC curve [17]. Indicative results of the Universal MASC curve are illustrated in Figure 5. The Universal MASC curve is useful for the precise residual stress measurement in an unknown type of steel: measuring the stress–strain dependence of a dog-bone of the unknown type of steel, with simultaneous measurement of MBN or magnetic permeability, results in the determination of the yield point and the maximum amount of the magnetic property, respectively. Thus, multiplying the Universal MASC curve with these values results in the determination of the MASC curve of the unknown type of steel [18].

In order to provide robust stress distribution monitoring, we need to have precise sensors. MBN measurements, if not resulting from coils surrounding the under-test steel, are subject to uncertainties due to the not-perfect vertical orientation of the sensor with respect to the under-test steel. A slight deviation from their vertical position with respect to the surface of the steel under test, lower than a degree, results in significant changes of the MBN signal, more than 10%. Apart from that, MBN measurements suffer from the rather random, non-deterministic and non-repeatable MBN, requiring a sequence of measurements at each point to average the response and therefore result in a less uncertain measurement. Magnetic permeability measurements suffer less from this effect, being proportional to the integration of the MBN with respect to time or field. For this reason, the development of new types of robust permeability sensors that are able to provide precise information on the permeability of the under-test steel are required. The families of permeability-measuring instruments and sensors, developed in the laboratory, are presented next, illustrating their uncertainty levels and robust response. At the first page, the single sheet tester of the lab is presented, followed by the electromagnetic yoke sensors, both being principles of commonly accepted methods for permeability measurements in several labs around the world. Then, two other types of permeability sensors are presented based on the Hall effect and the anisotropic magnetoresistance (AMR) principle, offering several advantages related to in-field measurements.

## 3. Single Sheet Testers

The historical and reference instrumentation to monitor the magnetic permeability and the magnetization loop of soft magnetic materials has been the Epstein method [19], mainly dedicated to thin electric steel laminas. A large amount of these laminas is packed, arranged in a rectangular frame, thus allowing for a closed magnetic loop, permitting the application of Ampere’s law when an excitation and search coil are arranged around them. In the Epstein topology, the excitation and search coils are coupled by the closed magnetic loop of the orthogonal frame generated by the thin steel laminas. The advantage of the Epstein method is the easily applicable equations of Ampere’s law, resulting in a high signal to noise ratio in the order of 10^4^ and in relatively low uncertainty, because of the large amount of magnetic mass involved in the measurement. This advantage is also responsible for the main disadvantage of the method: with the Epstein method, the average of the magnetization loop of all packed laminas is available, not permitting the measurement of each individual piece of steel. For this reason, the international steel community commonly accepted the ISO-ASTM A1036-04 (2020) standard entitled ‘Standard Guide for Measuring Power Frequency Magnetic Properties of Flat-Rolled Electrical Steels Using Small Single Sheet Testers’ [20], that allows for the determination of the dependence of the differential permeability and magnetization loop on the applied magnetic field using the so-called single sheet tester (SST).

We have adopted this standard and the experimental arrangement used in our lab is illustrated in Figure 6. With this arrangement, it is possible to measure the permeability and magnetization loop of different parts of a single steel sheet in low frequencies to avoid eddy currents in it. In fact, the excitation frequencies are from 0.1 Hz up to 10 Hz. In this arrangement, the excitation (or primary) and search (or secondary) coils surround the steel sheet under test and are directly coupled together. Our single sheet tester has been designed to accommodate single sheets with a thickness up to 0.5 mm and width up to 30 mm. Therefore, the constraints of the substrate of the primary and secondary coils were to have an internal free space of 1 mm and 31 mm external cross-section dimensions of 2 and 32 mm, respectively. The need to apply the Induction Law without compromises resulted in designing and developing a single-layer solenoid as the secondary coil. This coil should be short in order to monitor a small volume of the steel sheet under test. Therefore, it has been designed to be 10 mm long, made from 0.05 mm enameled copper wire, thus allowing for 200 single-turn secondary solenoid. The primary coil should be longer than the secondary coil in order to apply a uniform field inside the secondary coil, so designed to be 30 mm long. Therefore, the substrate of the primary and secondary coils was 32 mm long with an internal 0.5 mm deep gap of 10 mm length in the middle of the substrate to accommodate the secondary coil, made by polymer powder using a 3D printer. The substrate was polished with alcohol in order to have a smooth surface allowing for no complexities of the secondary coil wire winding.

The low excitation frequency permits the use of multiple turns of the excitation coil. According to the needs of the biasing field, 0.3 mm up to 0.5 mm enameled copper wire were used for the development of the excitation coil. For 0.3 mm and 0.5 mm wire, 100 and 60 turns are accommodated in a single turn, respectively, thus requiring for three to four turns for the 0.3 mm wire and five to six turns for the 0.5 mm wire to make the excitation coil. Different excitation and search coils with their corresponding substrates can be developed in order to accommodate different cross-sections of the under-test steel.

The requirements of the ISO-ASTM A1036-04 (2020) standard [20] refer to the presence of two mirror-like, soft magnetic yokes that trap the magnetic flux allowing for the precise average length calculation on Ampere’s law. There are some arguments on the type of the material of these yokes. A group of scientists and engineers wish to employ yokes with the maximum possible differential permeability to trap all magnetic lines (flux) from the single sheet. Another group wishes to use the same steel grade with the lamina under test. However, all agree to use yokes with a width larger than the width of the steel sheet under test. We have decided to use a couple of yokes made of amorphous ribbons with high differential permeability and magnetostriction close to zero to allow for magnetic flux trapping, without allowing magnetic lines to move outside the yoke. The width of the yokes is 50 mm, larger than the maximum width of the steel sheet under that which is 30 mm.

The measurement of the applied field H can be performed by calculations based on the number of turns of the excitation coil and the amplitude of the transmitted current. However, we prefer to use Hall sensors at different and well-determined positions above the excitation coil. It is known that the dependence of the applied magnetic field follows a parabolic law on the distance from the surface of the steel [21]. Therefore, setting three Hall sensors on a substrate vertical to the steel under test, the parabolic projection of the field to the zero position results in the actual magnetic field H being applied on the single sheet. For this reason, four-pin Hall sensors from the market were arranged with a 5 mm distance from each other, with the closest Hall sensor been 1 mm from the surface of the primary coil. Thus, the magnetic field from each Hall sensor was measured for different excitation currents. The response of the Hall sensors for every excitation current had its own coefficients in the parabolic response. However, dividing these coefficients with the excitation current I resulted in collapsing all parabolic curves into a single one with an uncertainty of ±100 ppm. Thus, the parabolic response of the magnetic field at different positions x is given by:(3)$$H\left(x,\;I\right)=aIx^{2}+bIx+cI$$
where $c=10\;m^{-1}+10^{-3}m^{-1}$. Thus, the magnetic field on the surface (and therefore in the bulk) of the steel sheet is given by:(4)$$H\left(I\right)=cI$$

During measurements, the magnetic field was determined just by a measuring the excitation current I. Measuring B(H) in the arrangement of Figure 6 can be achieved by using Ampere’s law:(5)$$\frac{N_{e}I(t)}{\sum \frac{1}{\mu_{i}(t)}\frac{l_{i}}{S_{i}}}=\Phi \left(t\right)$$
where $N_{e},\;{\;\mu }_{i,\;}\;l_{i},\;{\;S}_{i}\;and\;\Phi \left(t\right)$ are the number of turns of the primary coil, the differential permeability at a given excitation field at a given part of the magnetic circuit, the average length of the magnetic part, the cross-section of the magnetic part and the flux passing through the magnetic circuit, respectively. So, for the case of the single sheet tester illustrated in Figure 6, Ampere’s law is(6)$$\frac{N_{e}I(t)}{\frac{1}{\mu_{sss}(t)}\frac{l_{sss}}{S_{sss}}+2\frac{1}{\mu_{y}(t)}\frac{l_{y}}{S_{y}}}=\Phi_{sss}\left(t\right)$$
where $\mu_{sss}\left(t\right),\;l_{sss},\;S_{sss}$ are the differential magnetic permeability, the length and the cross-section of the single sheet tester, $\mu_{y}\left(t\right),\;l_{y},\;S_{y}$ are the differential magnetic permeability, the length and the cross-section of the yokes and $N_{e}$ is the number of turns of the primary coil. The lengths and cross-sections can be determined with relatively low uncertainty. The $\mu_{y}$ is a parameter that can be determined by using two pairs of yokes: each one of the yokes is fed by a primary (excitation) and a secondary (search) coil, having the same characteristics like the single sheet tester in terms of number of turns, length and diameter of the enameled copper wire. Then, these two yokes are set opposite each other with one exciting the other measuring the output signal. Ampere’s law for this case is(7)$$\frac{N_{e}I(t)}{2\frac{1}{\mu_{y}(t)}\frac{l_{y}}{S_{y}}}=\Phi_{y}\left(t\right)$$

Hence,(8)$$\mu_{y}(t)=2\frac{l_{y}}{S_{y}}\frac{\Phi_{y}\left(t\right)}{N_{e}I(t)}$$
but Φ(t) is related to the voltage output of the secondary (search) coil:(9)$$V\left(t\right)=-N_{o}\frac{d\Phi }{dt}$$

Therefore,(10)$$\Phi \left(t\right)=-\frac{1}{N_{o}}\int V_{y}\left(t\right)dt$$

So,(11)$$\mu_{y}\left(t\right)=2\frac{l_{y}}{S_{y}}\frac{\int V_{y}\left(t\right)dt}{N_{e}N_{o}I\left(t\right)}$$

Or(12)$$\mu_{y}\left(I\right)=2\frac{l_{y}}{S_{y}}\frac{\left(\int V_{y}\left(t\right)dt|I\right)}{N_{e}N_{o}I}$$

Thus, the μ_sss_, which is the actual unknown parameter, using Equation (6), is given by(13)$$\frac{N_{e}I}{\frac{1}{\mu_{sss}(I)}\frac{l_{sss}}{S_{sss}}+2\frac{1}{2\frac{l_{y}}{S_{y}}\frac{\left(\int V_{y}\left(t\right)dt|I\right)}{N_{e}N_{o}I}}\frac{l_{y}}{S_{y}}}=\Phi_{sss}\left(I\right)=-\frac{1}{N_{o}}\int \left(V_{sss}\left(t\right)dt|I\right)$$

Or(14)$$\mu_{sss}\left(I\right)=\frac{1}{N_{o}N_{e}I}\frac{l_{sss}}{S_{sss}}\frac{\left(\int V_{y}\left(t\right)dt|I\right)\left(\int V_{sss}\left(t\right)dt|I\right)}{\left(\int V_{y}\left(t\right)dt|I\right)-\left(\int V_{sss}\left(t\right)dt|I\right)}$$

All parameters at the right part of Equation (14) are known or can be known. Therefore, $\mu_{e}(I)$ or the $\mu_{sss}(H)$ can be determined, since H=cI. This is the software logic to precisely determine $\mu_{sss}(H)$. Apart from that, there is also another hardware solution. According to this, two secondary coils are arranged symmetrically around the axis of the primary coil, while one is empty and the other includes the steel under test. The two coils are connected in series opposition and their output voltage is proportional to the susceptibility of the steel sheet, instead of the permeability, like in the previous arrangement. The errors introduced by the fact that the cross-section of the secondary coil(s) is not the same with the steel sheet is rather negligible since almost all magnetic flux is included in the steel sheet. This way, a steel coupon with uniform distribution of residual stresses can be calibrated in terms of magnetic properties. The nature of this measurement methodology, i.e., the fact that a piece of steel is cut to be characterized, indicates that the single sheet tester cannot be used for field measurements.

Despite its disability of being used in the field, SST can be used for several lab measurements, determining the quality of steels, monitoring frequency dependence, determine minor loops and therefore determine the first-order reversal curve (FORC). Knowing the characteristics of the magnetization process, a number of parameters can be tested this way. Furthermore, the SST arrangement can also be used for MBN measurements in a way that the uncertainties caused by the precision of the vertical arrangement of the MBN search coil are missing in the SST arrangement: the search coil is almost impossible to move in an angle with respect to the sample orientation. Having the disadvantage of not being a really localized measurement, the SST arrangement used for MBN measurements is the proper MBN set-up allowing for small measurement uncertainties.

## 4. Electromagnetic Yokes

The electromagnetic yoke (EMY) employs some principles of the single sheet tester standard and is capable of being used in this field, since it can scan surfaces without any need of cutting the steel under test. The Π shape of these electromagnetic yokes allows for the yoke magnetization using a low-frequency current. Typical arrangements of electromagnetic yokes are illustrated in Figure 7, where the schematic of such a typical yoke is demonstrated with an excitation coil transmitting a low-frequency sinusoidal current, magnetizing the yoke. The search coil is also on the Π-shaped yoke, where the proximity with the steel under test offers a closed magnetic loop, magnetizing the steel under test, thus allowing the calculation of its permeability by using Ampere’s law. An important issue of this measurement is the distance between the yoke and the under-test steel. This gap may be of significant importance, being decisive for the uncertainty of the permeability measurement. For this reason, a contactless sensor for low air-gap determination, namely time-of-flight ultrasonic or laser transceiver, can be used.

At first, the permeability dependence on magnetic field is the same as for the case of the calibration of yokes of the single sheet tester. Thus, Equation (12) can be used to determine the permeability dependence on current and excitation field. Then, following the corresponding procedure for the single sheet tester, Ampere’s law for the case of a zero gap between the yoke and steel under test gives(15)$$\mu_{s}(I)=\frac{1}{N_{o}N_{e}I}\frac{l_{s}}{S_{s}}\frac{\left(\int V_{y}\left(t\right)dt|I\right)\left(\int V_{s}\left(t\right)dt|I\right)}{\left(\int V_{y}\left(t\right)dt|I\right)-\left(\int V_{s}\left(t\right)dt|I\right)}$$
where $\mu_{s}$ is the permeability of the steel under test and $V_{s}(t)$ is the voltage output of the search coil of the yoke during measurement of the steel under test. Taking the air gap into account, Equation (15) is more complicated, but can be analytically solved, too. In case the electromagnetic yoke is to be used in curly surfaces, its legs are of circular shape, allowing for continuous touching on the surface of the steel under test. However, in this case, the precision of measurement is not as good as for the case of flat yoke legs, because magnetic flux escapes from the curly surface of the yoke in a rather unpredictable way. In electromagnetic yokes, the question of the kind of the yoke’s material is also present. The two different schools are also present: use of either high-permeability material, or material of the same steel grade as the steel under test. In fact, in our group, we used highly permeable amorphous ribbon yoke, while high-quality electronics can clearly distinguish the change of the signal due to the presence of the steel under test. Electronics refer to current power amplifiers, like IGBT transistors and high-quality signal amplifiers together with 16-bit A to D converters, controlled by a proper micro-controller, like ESP32. Apart from that, the measurement of the ambient temperature is also required, since permeability is strongly dependent on temperature. Thus, both lift-off between yoke and steel under test, as well as the ambient temperature are two significant parameters, affecting the uncertainty of measurements. Precise measurements of permeability require measurements of both parameters (lift-off and ambient temperature) in order to correct the readout of the search coil of the yoke. This is important when in-field measurements are considered.

Although electromagnetic yokes are used in steel non-destructive tests, they suffer from several disadvantages. A certain disadvantage is the large area in which the yoke is able to measure permeability, while the other is the time needed for each individual measurement, being at least equal to the period of the excitation current. In case of 0.1 Hz, the time needed for a single measurement in a single direction is more than 10 s, which is very long for specific applications. This way, when electromagnetic yokes need to be used in the field, their excitation frequency is larger, up to 10 Hz, to allow for faster measurements. Such a frequency increase results in eddy currents involved in the output signal which is not favorable for a proper quasi-dc permeability measurement.

The way of using electromagnetic yokes in real environments depends on the specific in-field applications. In case that electric energy for feeding the electronics of the yoke and particularly the current amplifier of the excitation coil are acceptable for field measurements, electromagnetic yokes can be used, offering a large amount of information on magnetic properties. However, there are cases where energy requirements do not permit the use of the electromagnetic yokes. One reason may be that such energy is not readily available in the field, while the other is that the sensor must consume energy up to a certain threshold for anti-explosive reasons. For both cases, electromagnetic yokes cannot be efficient. Bearing in mind that at least two A rms are required for a proper permeability measurement, under at least 5 V power supply, the total energy consumption of the yoke is larger than 10 W, thus making its use impossible in cases of a restricted amount of energy demand.

However, electromagnetic yokes can offer significant information in quality control labs, where the speed of measurement and energy consumption are not an issue, because they offer information that may approach the information and the quality of information provided by the single sheet testers. As an example, the use of these yokes in ductile to brittle transition temperature in quality control labs of the steel industry is of vital importance, since they can offer detailed information of FORC analysis of magnetization loops. For some particular applications of steel structures, like heat exchangers, the use of electromagnetic yokes is preferable with respect to single sheet testers, due to shape reasons and the disability of cutting them for quality control.

For all these reasons, we have developed new families of sensors, having a much smaller size, much lower energy consumption, having a time response in the order of ms or even μs, allowing for fast, reliable and safe measurements, with power consumption in the order of mW. These families of sensors are based on the Hall effect and the Anistropic Magnetoresistance (AMR) effect and are presented next.

## 5. Hall Sensors

The Hall effect is well-known for sensors and transducers [22]. They offer a linear dependence of the magnetic field component vertical to the surface of the sensing element, being able to have a large variety of span from mT up to T, with a sensitivity in the order of μTHz^−1/2^. This sensitivity requires the magnetic biasing field to properly use the advantages of Hall sensors in magnetic permeability measurements. Therefore, the design of our permeability sensors based on the Hall effect use the principle of magnetic bias. Suppose a permanent magnet yoke, i.e., a yoke comprised of a permanent magnet, magnetizing two magnetized soft magnetic bars, as depicted in Figure 8, comprising the so-called permanent magnet yoke. In the absence of a magnetic steel under test, magnetic lines follow a large path from one soft magnetic pole to the other (left). As soon as the magnetic steel is under the permanent magnet yoke at a certain distance from the soft magnetic bars, the magnetic lines tend to be more vertical towards the surface of the magnetic steel under test (middle). The larger the permeability of the steel under test, the more vertical the magnetic lines on the surface of the steel, provided that the distance between the soft magnetic bars and the steel under test remains the same. Therefore, the magnetic field at the edge of the soft magnetic bar increases with the permeability of the steel under test.

At the same time, the magnetic field far from the edge of the soft magnetic bar decreases with the permeability of the magnetic steel. It is clear that for this family of sensors, the measurement of the distance between the soft magnetic bar and the steel under test is of vital importance. Similarly, the ambient temperature is also important, affecting both, the performance of the Hall sensor and the permeability of the steel under test. Therefore, a time-of-flight sensor and an ambient temperature sensor are used between the two soft magnetic bars to adjust the Hall sensor response at the proper levels, thus affecting dramatically the uncertainty of the measurement. In Figure 8, the actual permanent magnet yoke with the Hall sensors at the bottom of the soft magnetic bars can be seen. The packaging of the system, including a 16-bit A to D converter, together with an ESP32 microcontroller, inside a 3D-printed package, can also be observed (right). This sensor can also measure magnetic anisotropy by rotating it on top of the steel under test. Such a measurement is illustrated in Figure 9.

Apart from this type of Hall sensor arrangement, there is another possible magnetization means, comprised of a permanent magnet in the form of a cylinder, being vertical to the surface of the steel under test, with its magnetization along the axis of the cylinder. Thus, following the same principle like in Figure 8, in the absence of steel under test, the magnetic lines getting out of the cylinder follow a large path until entering back to the other pole of the permanent magnetic cylinder. In the presence of the steel under test, the magnetic lines are more vertical on the surface of the steel, dependent on the local permeability and magnetic anisotropy of the steel, while they escape from the steel at a distance also dependent on the local permeability and magnetic anisotropy of the steel.

Therefore, arranging a ring of Hall sensors, radially oriented and surrounding the edge of the permanent magnetic cylinder, results in the measurement of the local magnetic permeability of the steel on a ring, formulated by the diameter of the permanent magnetic cylinder. The distribution of permeability is the indication of the magnetic anisotropy of this region of the steel under test. A typical response of such anisotropy is indicated in Figure 10, using a 12 mm × 12 mm Si substrate, where 24 Hall sensors radially developed with a 15° pitch, having a 10 mm diameter permanent magnetic bar in the middle of the arrangement, are used.

The advantages of this sensor arrangement open a new era in magnetic permeability measurements in steels. In the first place, they can achieve a fast measurement. Bearing in mind that the time response of the Hall sensor, including the A to D conversion and signal transmission through the microcontroller of the sensor is less than 1 ms, and the fact that the size of the Hall sensors is in the order of 0.1 mm × 0.1 mm, the speed of measurement is in the order of 0.1 m/s or 360 m/h with a spatial resolution of 0.1 mm. This speed is sufficient for several steel production and manufacturing lines. Furthermore, it is incomparably higher than the speed of electromagnetic yokes. Furthermore, using the permanent magnetic cylinder arrangement, the magnetic anisotropy is able to be monitored with the same speed and spatial resolution. Apart from that, the power consumption of the permanent magnet yoke is in the order of 50 mA, including the ESP32 microcontroller, allowing for the possibility of passing the anti-explosive ATEX specifications [23]. However, the permanent yoke arrangement suffers from the same size issues, like the electromagnetic yoke, due to the similar topology of the Π-shaped yoke. In order to improve sensitivity, energy consumption and other properties, another family of permeability sensors has been developed, based on the AMR effect, presented in the next chapter.

## 6. AMR Sensors

AMR sensors are more sensitive than the Hall sensors. In fact, they can have a sensitivity in the order of 1 nTHz^−1/2^ and a span of several hundreds of μT. This way, they do not need a biasing magnetic field in the order of several tens of mT to operate as the Hall sensors do. Instead, they can operate with the biasing field of the Earth’s field, being in the order of some tens of μT. This way, they do not need permanent magnets to measure permeability. Although they are used as sensing elements in electromagnetic and magnetic yokes to monitor cracks present on the surface of the steel, they can also be used without these yokes to monitor the magnetic permeability in given surface areas of the steel under test, having a size in the order of the size of the AMR sensor.

The main principle of operation is the following: suppose that the steel under test has a residual stress on its surface. If this material is positive magnetostrictive in the direction of measurement and has a tensile stress field in this direction, then its permeability increases [24]. Bearing in mind that the Earth’s field is parallel to the surface of the steel and having such a local permeability increase, the Earth’s field (magnetic lines) penetrates the steel under test in the region of the increased permeability. Earth’s field has three components: $H_{x}$ corresponding to the in-plane direction along measurement (X-axis), $H_{y}$ corresponding to the in-plane direction orthogonal to the measurement axis (Y-axis) and $H_{z}$ corresponding to the direction vertical to the plane of measurement (Z-axis). Then, $H_{z}$ will decrease (the Z-axis is considered positive for amplitudes from the surface of the steel under test upwards) and therefore $H_{x}$ will slightly decrease, because the magnetic lines of the X-axis are trapped inside the steel. Consequently, the $H_{y}$ will slightly increase, because it has to behave the opposite of the X-axis, because the sign of magnetostriction should be opposite to the sign of magnetostriction of the X-axis [25].

If the steel is positive magnetostrictive in the direction of measurement and has a compressive stress field in this direction, then its permeability decreases. Thus, following the same analysis as above, $H_{z}$ will increase and therefore $H_{x}$ will slightly increase, while $H_{y}$ will slightly decrease. If the steel is negative magnetostrictive in the direction of measurement and has a tensile stress field in this direction, then its permeability decreases. Following the same analysis as above, $H_{z}$ will increase and therefore $H_{x}$ will slightly increase, while $H_{y}$ will slightly decrease. If the steel is negative magnetostrictive in the direction of measurement and has a compressive stress field in this direction, then its permeability increases. Thus, $H_{z}$ will decrease and therefore $H_{x}$ will slightly decrease, while $H_{y}$ will slightly increase. All these are depicted in Table 1. A schematic describing this process is illustrated in Figure 11.

A critical issue arising from this analysis is how to distinguish positive and negative magnetostrictive materials in known and unknown steel grades in order to select the proper sign of residual stresses at the area of measurement. Regardless of the steel grade, a magnetostriction constant is not universally either positive or negative. It may be positive or negative in given crystallographic directions, but it is also dependent on the amount of magnetostriction in given coordinates: in case λ_x_ in the X-axis is positive at a given region, then λ_y_ in the Y-axis is negative in most cases at the same region. Until now, this issue has remained unsolved, and there is ongoing research in order to find a solution. However, since the gradient of stress is the important parameter, one should count on the absolute value of stress which appears to be symmetric with respect to the Y-axis of the MASC curve. Therefore, AMR scanning may provide the absolute value of stress along the measurement, which tells about the stress gradient. Apart from that, the areas of severe probability of suffering a nano-crack generation are those demonstrating a gradient from one sign of stress to another, when stresses are close to the yield point of the steel under test. Therefore, the absolute stress value and the gradient stress are the two decisive parameters for the determination of the position of crack initiation.

The AMR sensor used in our group is the ST microelectronics LSM30-3D sensor. It has been selected due to the sensitivity of 1 nTHz^−1/2^, the three-dimensional measurement of field and the ambient temperature sensor hosted on the integrated circuit, which is vital for more precise measurements. Two types of AMR sensors have been developed, one with a biasing permanent magnet and another one without permanent magnet yoke. A typical response of the AMR sensor on non-oriented electric steel which suffered induction heating and consequent quenching, therefore giving rise to residual stresses and strains, is illustrated in Figure 12. In this response, the coincidence of the sign of the AMR response in X and Z axes can be seen, as well as the opposite sign in the Y-axis direction. This response refers to the AMR sensor with the permanent magnet yoke. The same response without the yoke is illustrated in Figure 12 demonstrating better sensitivity, allowing for the monitoring of smaller residual stresses present along the length of the measurement.

In case the AMR sensor needs a lift-off for measurement, like a (non-magnetic) coating, a time-of-flight sensor, either ultrasonic or laser, can be used in order to correct the readout response. Otherwise, concerning naked steels under test, the polymeric sensor packaging is attaching the steel under test, taking into account the precise distance between the AMR sensor and the outer part of the sensor packaging, which is 0.5 mm ± 0.01 mm. The excitation and detection electronics are driven by the ESP32 microcontroller, used in all our Hall and AMR sensors arrangements. The main advantage of the AMR sensor is the ability to monitor three-dimensional fields with a sensitivity better than the Hall sensor. Another advantage of the AMR sensor is the fast measurement, less than 1 ms, including the time needed for the microcontroller. Bearing in mind the size of the AMR sensor being 0.1 mm × 0.1 mm, similar to the Hall sensor, it can scan surfaces as fast as the Hall sensor. The power consumption is also similar to the Hall sensor, with the most amount of power required for the microcontroller. This way, the AMR sensor can also be used in harsh environments as with the Hall sensor, provided that both pass the specific anti-explosive ATEX requirements.

Magnetic field sensors can in general be used for such measurements, like GMR [26], TMR [27], Spin Valves [28], and GMI [29,30,31,32] sensors. All these sensors can take the place of AMR sensors with similar results. Furthermore, other types of magnetic sensors can also be employed with the possibility of measuring residual stresses [33,34,35], using the magnetoelastic coupling theory [36,37].

Bearing in mind the advantages of the AMR sensor with respect to the rest of the sensors, we have provided the dependence of the magnetic component $H_{z}$ on the residual stresses in two different types of steels, namely non-oriented electric steel and 42CrMo4 steel grade, which is suitable for maritime applications. The two samples have been prepared by using localized RF induction heating and consequent quenching in water. The samples were measured in three different lines across the area of heating and quenching. The results are given in Figure 13, illustrating a sigmoid response, reaching values of residual stresses closer to the yield point with respect to the stress values obtained with the autogenous welding process.

## 7. Discussion

The four different families of instruments and sensors, able to measure and monitor the differential magnetic permeability of steels, are important in the steel industry, in both production and manufacturing, as well as in structural and operational steel applications. At first, the single sheet tester (SST) arrangement is useful for quality control laboratories in the steel industry. As an example, our SST instrument has been used in the quality control laboratory of Corinth Pipe Works (CPW) in Greece to monitor the ductile to brittle transition temperature (DBTT) in magnetic steels for pipeline manufacturing. The classic method of monitoring such transition temperature is to drop a weight from a distance and determine the energy required for breaking the steel. CPW accepted the use of our magnetic technology in their factory to monitor the permeability of such samples. Indeed, the experimental set-up was developed the quality control facilities of CPW, to compare the magnetic technique with the classical, standardized response of the drop weight facility of the factory. Next to this drop weight instrument, there is a cooling pool to drop the temperature of the steel under test. As soon as the temperature reaches the required level, the piece of steel is transferred quickly from the cooling pool to the drop weight instrument to drop the weight and then to determine the energy required for breaking the steel sample. As soon as the temperature of the steel is below DBTT, the breaking energy decreases. The SST instrumentation was set next to the above-mentioned cooling pool, and the magnetic permeability of the steel under test was determined at each temperature level. Figure 14 illustrates the set-up and the achieved results. According to these results, the ductile to brittle transition temperature window was within the region measured by the standard method. However, the time required to determine DBTT in one sample using the standard method is more than 8 h, while the magnetic method requires no more than 10 min for one sample, without any kind of special and laborious sample preparation. This application of the permeability sensors to determine DBTT, as well as to predict the crack initiation of magnetic steels are the two examples according to which the sensors and the methodology of using them to predict the crack initiation are clear indications of the superiority of these sensors with respect to the current state of the art.

The electromagnetic yoke has been used for the same reason, i.e., quality control procedures in testing magnetic steels, before the development of our low-power-consumption Hall and AMR sensors.

As already mentioned in previous chapters, there are two parameters affecting the performance of the permeability sensors. The first is the lift-off effect, causing uncertainties in measurements: an increase of the lift-off results in a decreased amplitude of the under-determination magnetic permeability, illustrating an incorrect amount of residual strain and stress at the area of measurement, according to the sigmoid response of the MASC curves presented in Section 2. The second is the ambient temperature during measurement. The dependence of permeability on temperature is more or less decreasing and parabolic. Permeability decreases with temperature, affecting the determination of residual stresses on the under-test steel too. The response of all permeability sensors is independent of the specific and mentioned application, like energy, transportation, oil and gas, steel structures, etc. The only parameters affecting their response are the lift-off effect and ambient temperature.

The steel applications foreseen by our laboratory are many and different. Apart from steel manufacturers (namely pipeline, shipyards, steel coil producers, heat exchanger manufacturers, etc.) where the inspection of the incoming raw materials and the final product are needed, the steel end-users that need such measurements are divided into five main categories:Transportation, namely on-board stress tests in ships, railways and train wagons, as well as electric steels for the largely developing industry of electric motors for e-motion applications, etc.Energy, namely heat exchangers in several energy production activities, test of domes in nuclear stations, tests of the carrying steel bar in underwater current transmission, etc.Oil and gas, namely quality control of the front-end tools to cut the ores, tests in vessels under pressure and pipelines of refineries, oil and gas transmission pipelines, etc.Several steel structures, like steel bridges, metallic buildings, special facilities like the 27 km ring and the foreseen 100 km new ring at CERN, etc.

The comparison of the four different families of sensors is depicted in Table 2, allowing for the determination of the advantages and disadvantages of each one. The Hall and AMR sensors can be used in harsh environments, like oil and gas transmission lines, vessels under pressure, etc., during operation and not only in the shut-down periods. Of course, it is impossible for them to be used in neutron fluence environments, like in nuclear stations. For this particular reason, new types of Hall and AMR sensors should be developed, made of semiconductors able to withstand high energies. Examples of these materials are the garnets that have been extensively used in space applications, having proven their resistance in high-energy particles. Apart from that, the AMR sensor can be used to develop a pen-like sensor for some specific applications: for example, surveyors in ships, energy stations, oil and gas applications and other steel structures may have such a pen, calibrated against residual stresses and strains, transferring data to their cell phone or tablet or laptop in order to see by their own eyes the validity of measurements and evaluate the quality control procedures, based on our magnetic steel health-monitoring technology.

Another important issue is the development of technology for the structural health monitoring of non-magnetic ferrous steels, like the austenitic 304, 316, 316L and 316LN steel grades. In fact, all these steels are non-magnetic until the end of their life: as soon as they suffer the development of nano-cracks, they undergo a phase transformation from austenitic to martensitic phase, which is magnetic. To demonstrate a feeling on such phase transformation, one can think of the following experiment: cutting an austenitic steel without cooling results in a magnetic layer at the vicinity of cutting, which corresponds to the martensitic phase. Thus, the measurement of a small magnetic substance, by the Hall or the AMR sensor, is an indication of the beginning of the end of life of the austenitic steel. As the sensitivity of the AMR sensor is superior to the Hall sensor, it is very logical to use AMR sensors for the austenitic steel health monitoring. Bearing in mind the vast amount of applications of austenitic steels, our team has initiated work in this subject, mainly targeting applications in ITER, south France.

One of the most significant future outcomes of magnetic steel health monitoring is the remaining lifetime prediction, based on the monitoring of the differential permeability change with time. The most straightforward method to foresee the position and the time of failure of such a steel grade is the projection of a continuous time-dependent function towards the yield point of the steel. The time-based projection is separated into micro-time, mid-time and macro-time functions of permeability change, representing the accumulation of residual and hydraulic stresses. For example, a sudden and short approach to the yield point due to a high hydraulic stress is not necessarily the preamble of a steel failure: if it happened once and then the magnetic properties are changing slightly, the steel in question does not suffer possible failure. Instead, a stress gradient changing from −20% up to 20% of the yield point in a space of a few mm will result in steel failure, in a time window determined by the projection of the actual data toward the yield point. Another approach apart from following a continuous projection of the magnetic permeability towards the yield point is the realization of high and low cycle fatigue studies (i.e., fatigue studies less or higher than the yield stress point, respectively), to have the experimental evidence of the permeability levels approaching the yield point.

Having determined the position and the time window for the steel failure, we are also able to fix the issue of residual stress by using a localized heat induction treatment. Transmitting a sinusoidal current at a frequency bandwidth from 10 to 30 kHz and a current amplitude in the order of 30 A results in eddy currents in the near-surface of the sample. Bearing in mind the low thermal conductivity of steel, it is expected that a temperature profile, with maximum temperature below one-third of the melting point of the corresponding steel grade, provides stress annihilation.

Future work includes experiments under low and high cycle fatigue under various thermodynamic and electromagnetic conditions.

The presented sensors and the technology to provide the correlation of the sensor response with the actual residual stresses in the tested steels is the basis, i.e., the working documents of the Quality Management System of our laboratory, towards its accreditation according to the ISO 17025 Standard.

## 8. Conclusions

In this work, after presenting the dependence of magnetic properties, like MBN and differential magnetic permeability, on residual stresses, the sensors measuring magnetic permeability were presented. These sensors are suitable for laboratory use, like the single sheet tester instrument or the electromagnetic yoke, as well as for in-field measurements, like sensors based on the Hall effect and the AMR principle. The advantages and disadvantages of these sensors have been discussed, offering applications in steel manufacturing and corresponding use in steel structures.

## Figures and Tables

**Figure 1 sensors-25-00606-f001:**
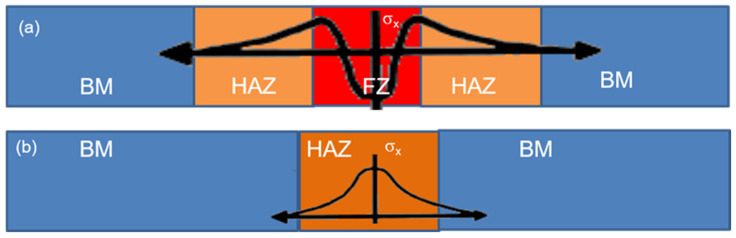
Stress distribution in steel coupons: (**a**) stresses in the fusion and heat-affected zones of welded samples; (**b**) stresses in inductively heated and quenched steel coupons.

**Figure 2 sensors-25-00606-f002:**
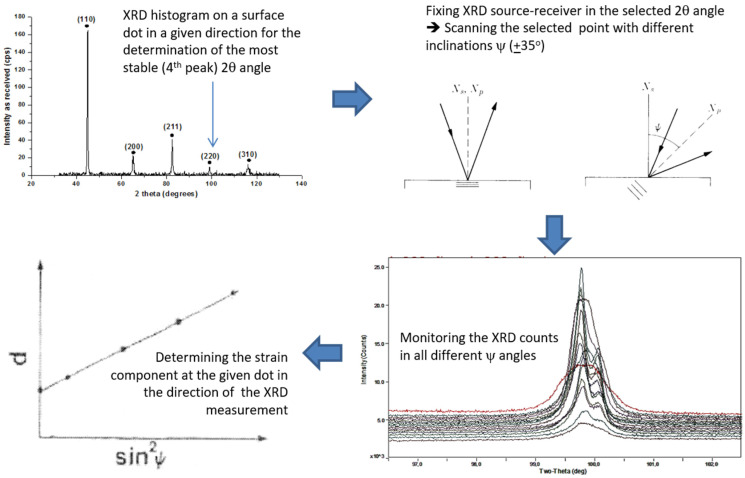
The methodology to determine the strain at a given dot of the under-test steel. At first, the XRD histogram determines the 2theta angle corresponding to the fourth (most stable) peak of the histogram (**top left**). Then, the X-ray source and detector are precisely fixed in this 2theta angle that rotates in an angle ±ψ (**top right**), allowing for the determination of counts with a signal to noise ratio of 10^4^ (**bottom right**). Finally, the dependence of counts d on ${sin}^{2}\psi$ determines the strain ε and stress σ (**bottom left**).

**Figure 3 sensors-25-00606-f003:**
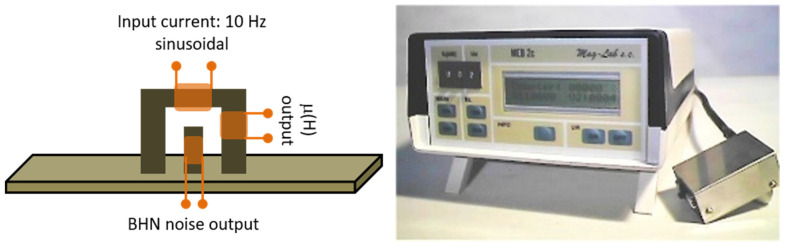
The MEB-2c probe used for the initial determination of magnetic properties. The schematic of the sensor (**left**). The actual probe with the electronics (**right**).

**Figure 4 sensors-25-00606-f004:**
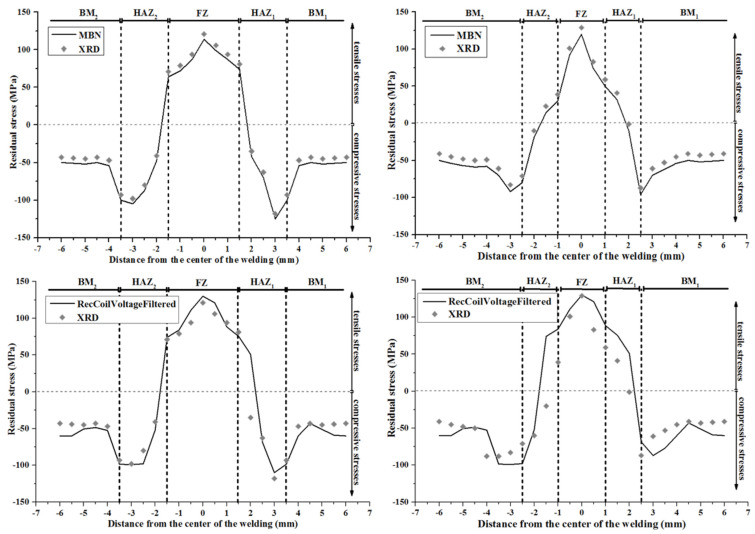
Indicative response of correlation between stresses and magnetic properties. **Top left**: correlation of MBN measurements with residual stresses in AISI 1008 steel, US Steel, Pittsburg, PA, USA, using TIG welding. **Top right**: correlation of MBN measurements with residual stresses in AISI 1008 steel using plasma welding. **Bottom left**: correlation of the differential permeability and stresses in AISI 1008 steel using TIG welding. **Bottom right**: correlation of the differential permeability and stresses in AISI 1008 steel using plasma welding. The larger uncertainty for the differential permeability response is due to the calibration of the electromagnetic yoke.

**Figure 5 sensors-25-00606-f005:**
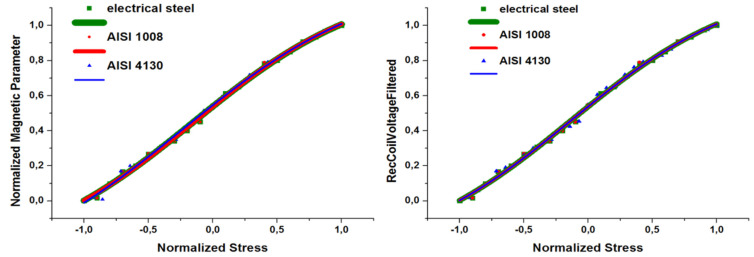
Universal MASC curves for MBN (**left**) and permeability measurements (**right**).

**Figure 6 sensors-25-00606-f006:**
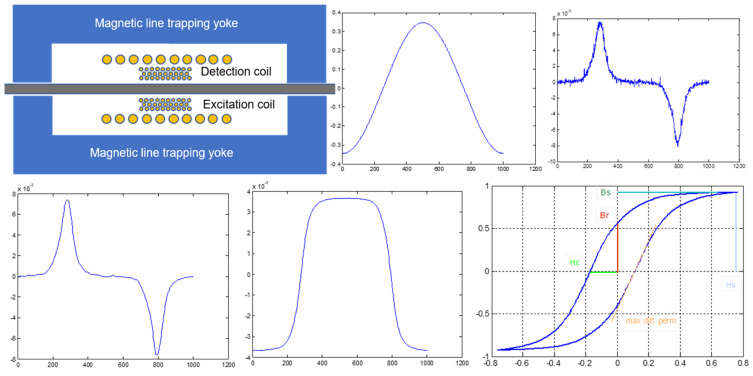
The schematic and waveforms of the single sheet tester of our lab. **Top left**: the schematic of the single sheet tester. **Top middle**: excitation current waveform proportional to the excitation field H. **Top right**: the voltage output of the secondary coil. **Bottom left**: the filtered voltage output of the secondary coil. **Bottom middle**: the integrated voltage output of the secondary coil, proportional to the magnetic flux B. **Bottom right**: the magnetization loop as a plot of B versus H.

**Figure 7 sensors-25-00606-f007:**
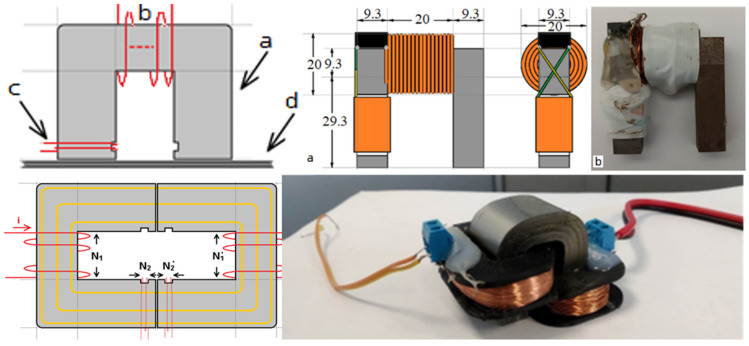
Typical electromagnetic yoke arrangements. **Top left**: schematic of the electromagnetic yoke (a), with the excitation coil (b), search coil (c) and steel under test (d). **Top right (a)**: exact geometrical configuration of an ARMCO yoke. **Top right (b)**: the actual ARMCO electromagnetic yoke. **Bottom left:** calibration methodology of the permeability of the electromagnetic yoke. **Bottom right**: near-zero magnetostriction amorphous ribbon yoke with the excitation and detection coils, able to operate on flat steel surfaces.

**Figure 8 sensors-25-00606-f008:**
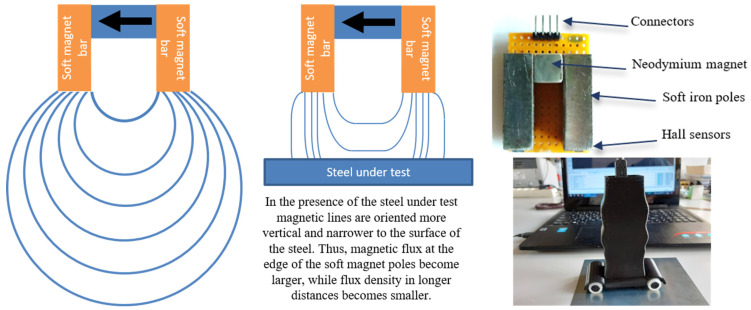
The main principle of our family of permeability sensors based on the Hall effect. **Left**: magnetic lines in the absence of the steel under test. **Middle**: magnetic lines in the presence of the steel under test. **Right**: the actual permanent magnet yoke with the Hall sensors at the bottom of the soft magnetic bars and the packaging of the Hall sensor arrangement, hosting four wheels allowing for permeability distribution monitoring.

**Figure 9 sensors-25-00606-f009:**
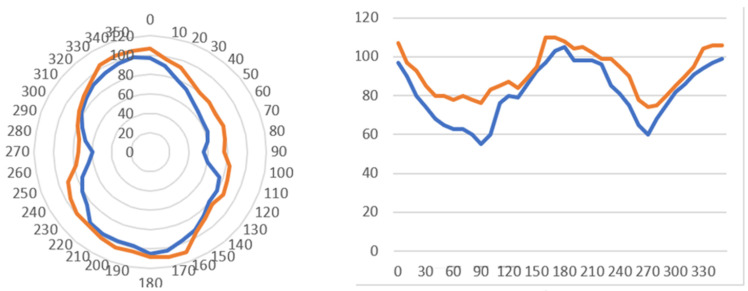
Magnetic anisotropy measurements, obtained by rotating the sensor on top of the steel under test in different areas of non-oriented electric steel. **Left**: Polar anisotropy response. **Right**: Cartesian anisotropy response.

**Figure 10 sensors-25-00606-f010:**
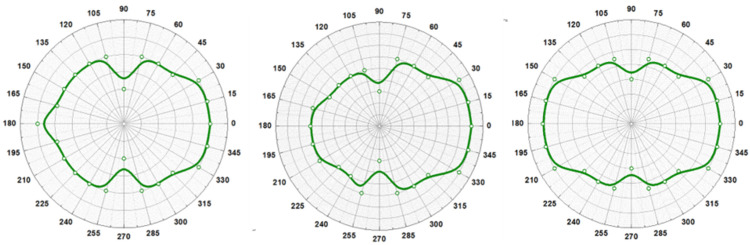
Polar magnetic anisotropy (arbitrary units) in the heat-affected zones (HAZ) of welded AISI 1008 flat steel samples. **Left**: TIG-welded sample. **Middle**: Plasma-welded sample. **Right**: Induction heating sample.

**Figure 11 sensors-25-00606-f011:**
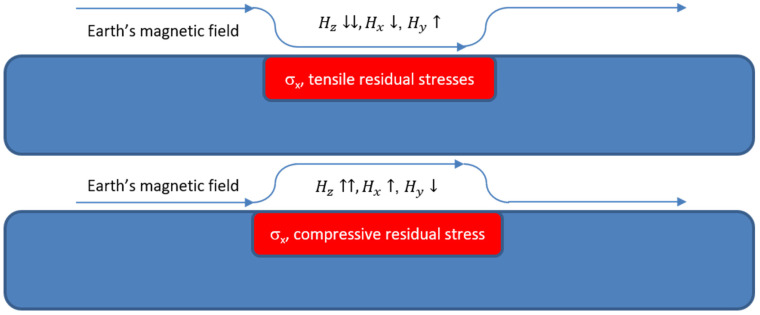
Representation of the magnetic lines of Earth’s field for tensile and compressive stresses in positive magnetostrictive steels. The opposite behavior occurs in negative magnetostrictive materials.

**Figure 12 sensors-25-00606-f012:**
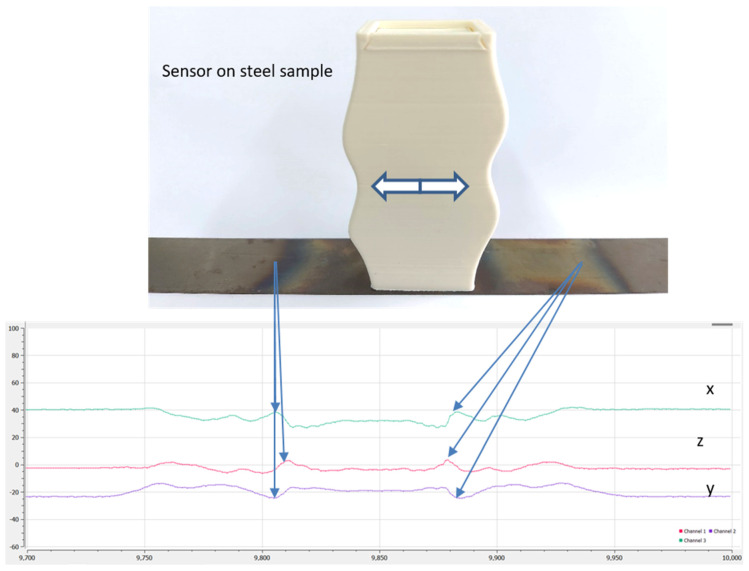
The AMR sensor used with biasing permanent magnet yoke, able to show the changes of field in the X, Y and Z axes, demonstrating the presence of stresses in a non-oriented steel after local induction heating and consequent quenching. **Top**: The sensing element from ST microelectronics, together with its packaging on top of a non-oriented steel. **Bottom**: Response of the field change on the three directions X, Y and Z.

**Figure 13 sensors-25-00606-f013:**
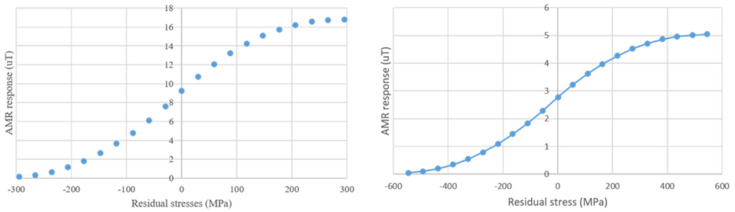
MASC curves of non-oriented electric steel (**left**) and 42CrMo4 maritime steel grade (**right**), using RF induction heating and consequent quenching.

**Figure 14 sensors-25-00606-f014:**
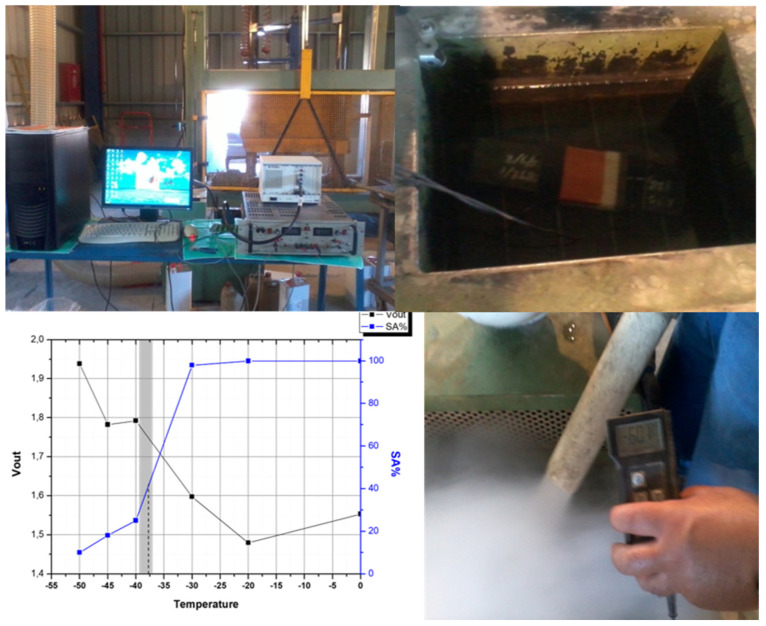
The SST of our lab used for the DBTT determination. Top left: the SST of our lab (behind SST a part of the standard Drop Weight test can be seen). Top right: the cooling pool where the electro-mechanical part of the SST instrument was firmly set. Bottom right: manual measurement of temperature for security reasons. Bottom left: comparison of the magnetic measurements with the drop weight tests. The agreement between the two methods can be seen.

**Table 1 sensors-25-00606-t001:** Change of permeability and AMR sensor response due to stresses and magnetostriction.

λ	σ	μ	$H_{z}$	$H_{x}$	$H_{y}$
+	+	↑	↓↓	↓	↑
+	−	↓	↑↑	↑	↓
−	+	↓	↑↑	↑	↓
−	−	↑	↓↓	↓	↑

**Table 2 sensors-25-00606-t002:** Comparison of the four families of permeability sensors.

Characteristics Sensors	Use in the Field	Consumption (W)	Speed of Tests (mm/s)	Sensitivity (ppm)	Uncertainty (%)	Cost of Use (k€)
SST	No	>100	0.01	10	0.1	50–180
EMY	Yes	>50	0.01	100	0.5	30–60
Hall	Yes	~0.05	100	1000	±2	0.5–1
AMR	Yes	~0.05	100	1000	±1	0.5–1

## Data Availability

Data are available at request by the author.

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
