# Peer review of "Permeability Sensors for Magnetic Steel Structural Health Monitoring"

_sensors, 2025, doi:10.3390/s25030606_

Round 1

Reviewer 1 Report

Comments and Suggestions for Authors

This paper presents the dependence of magnetic properties, like MBN and differential magnetic permeability, on residual stresses. However, I have some additional comments for the authors:

(1)    Most of the figures in the manuscript are not clear enough, for example, the graphs are blurred, I suggest asking the authors to improve the clarity of the figures before publishing them so that they can be easier for readers to read.

(2)    Subfigures in most figures need to be labelled and supplemented with caption.

(3)    In recent years, TMR sensors have also been widely used for magnetic field measurements, which are suggested to be supplemented and analyzed comparatively.

(4)    In experiments, it is suggested that the authors could give an application case of the proposed measurement method so as to verify the superiority to other traditional method.

Author Response

Dear Reviewer, thank you for your valuable comments. All of them have been into consideration and are answered in the attached file.

Reviewer 2 Report

Comments and Suggestions for Authors

The document aims to monitor the structural health of magnetic steel using magnetic permeability sensors and residual stress measurements.

The paper provides a detailed analysis of the behaviours of steel structures with the proposed type, but it has some limitations.

The document thoroughly discusses the behaviour of steels in the presence of environmental noise and extreme temperatures. Can the authors provide more detailed explanations on the behaviour of the proposed system in the presence of steels with variable characteristics, such as complex heat treatments or multiphase compositions?

The dependence of some measurements on the distance between the sensors and the surfaces: what errors could it generate in practical applications?

The same applies to the behaviour with changes in ambient temperature.The document lacks a detailed comparative evaluation of the relative effectiveness of the techniques in different sectors, such as energy compared to transportation or compared to emerging and established sectors like the biomedical field. For this reason, are there comparative studies that evaluate the sensitivity and accuracy of AMR and Hall sensors compared to competing technologies in industrial applications?

The authors evaluated the system's performance considering the implementation of machine learning algorithms to improve the correlation between magnetic measurements and residual stresses.It is worth noting that the processed signals can often be subject to uncertainties and/or inaccuracies that render the acquired measurements inefficient. In these cases, recent scientific literature suggests using techniques based on fuzzy logic. In particular, the most recent research guidelines recommend grouping similar data in a fuzzy sense (through fuzzy divergence calculations that prove to be a measure of distance in a particular functional space) into specific groups from which to extract, for each of them, a representative characteristic of each grouping. Then, the classification of the signal can be performed by comparing the obtained fuzzy divergences with each representative data of each grouping. Obviously, I don't expect the authors to implement such a tool. I only ask that a brief paragraph be included in the text highlighting this possibility by listing the following relevant article in the bibliography: doi: 10.3233/ICA-230730. 

The document describes the methods for sensor calibration, such as the use of steel coupons. Can the authors elaborate in detail on the reproducibility of measurements in different contexts?

The authors describe the use of sensors in difficult environments. The authors can provide a comparison with other data, preferably experimental, to demonstrate the robustness of the technologies in extreme conditions such as high humidity or prolonged exposure to intense magnetic fields?

Can the authors integrate the behaviour of the sensors in the presence of steels with complex chemical compositions or specific heat treatments into the document?

Finally, the authors could improve the quality of the figures, especially Figures 2 and 4.

Thank you.

Author Response

Dear Reviewer, thank you for your valuable comments. All of them have been into consideration. Please see the attached file.

Round 2

Reviewer 1 Report

Comments and Suggestions for Authors

The paper is well revised.

Author Response

Dear Reviewer, thank you for your comments

Reviewer 2 Report

Comments and Suggestions for Authors

The paper proposes to monitor the structural health of magnetic steel using magnetic permeability sensors and residual stress measurements. The paper provides a detailed analysis of the behaviour of steel structures with the proposed typology but has some limitations. The paper describes methods for the calibration of sensors, such as the use of steel coupons, in which the reproducibility of measurements in different contexts is described in detail. There are no particular comments on the proposed work.

Author Response

Dear Reviewer, thank you for your comments. The developed stress coupons with welding methodology, as well as induction heating and corresponding quenching are illustrated in the manuscript. The reasoning of these stress coupons has also been demonstrated.